# Protective Effects of Some Grapevine Polyphenols against Naturally Occurring Neuronal Death

**DOI:** 10.3390/molecules25122925

**Published:** 2020-06-25

**Authors:** Laura Lossi, Adalberto Merighi, Vittorino Novello, Alessandra Ferrandino

**Affiliations:** 1Department of Veterinary Sciences (DSV), University of Turin, 10095 Grugliasco (TO), Italy; adalberto.merighi@unito.it; 2Department of Agricultural, Forestry and Food Sciences (DISAFA), University of Turin, 10095 Grugliasco (TO), Italy; vittorino.novello@unito.it

**Keywords:** resveratrol, polydatin, peonidin 3-O-glucoside, malvidin 3-O-glucoside, quercetin 3-O-glucoside, (+)-catechin, (+)-taxifolin, apoptosis, neuronal death, cerebellum

## Abstract

The interest in the biological properties of grapevine polyphenols (PPs) in neuroprotection is continuously growing in the hope of finding translational applications. However, there are several concerns about the specificity of action of these molecules that appear to act non-specifically on the permeability of cellular membranes. Naturally occurring neuronal death (NOND) during cerebellar maturation is a well characterized postnatal event that is very useful to investigate the death and rescue of neurons. We here aimed to establish a baseline comparative study of the potential to counteract NOND of certain grapevine PPs of interest for the oenology. To do so, we tested ex vivo the neuroprotective activity of peonidin- and malvidin-3-O-glucosides, resveratrol, polydatin, quercetin-3-O-glucoside, (+)-taxifolin, and (+)-catechin. The addition of these molecules (50 μM) to organotypic cultures of mouse cerebellum explanted at postnatal day 7, when NOND reaches a physiological peak, resulted in statistically significant (two-tailed Mann–Whitney test—p < 0.001) reductions of the density of dead cells (propidium iodide^+^ cells/mm^2^) except for malvidin-3-O-glucoside. The stilbenes were less effective in reducing cell death (to 51–60%) in comparison to flavanols, (+)-taxifolin and quercetin 3-O-glucoside (to 69–72%). Thus, molecules with a -OH group in ortho position (taxifolin, quercetin 3-O-glucoside, (+)-catechin, and peonidin 3-O-glucoside) have a higher capability to limit death of cerebellar neurons. As NOND is apoptotic, we speculate that PPs act by inhibiting executioner caspase 3.

## 1. Introduction

We know a great deal about the chemistry of grapevine polyphenols (PPs) because they are widely studied for their implications in wine production and biological role in the grapevine response to biotic and abiotic stress. Notably, there is also a huge body of preclinical evidence on the numerous cellular mechanisms targeted by these substances (resveratrol, in particular) in relation to several pathological conditions including neurological diseases [1,2], but results are often heterogeneous or inconsistent. There may be several reasons for the heterogeneity and lack of consistency of in vitro and/or animal studies on (grapevine) PPs. First, these studies have come under heavy criticism because they have used artificially high doses. However, an additional and even more important concern is that they are unreliable because many of the effects of polyhydroxylated natural phytochemicals, such as resveratrol and epigallocatechin gallate, were reported to be due to aspecific cell membrane perturbations, rather than specific protein binding [3]. Therefore, doubts arose that these molecules are pan-assay interference compounds that affect the accuracy of many assays [4,5]. Other more particular reasons for the divergences in the outcomes of animal experiments may be the following. First, researchers at times have tested the effects of pure molecules but, other times, those of crude plant extracts, with a substantial heterogeneity of results that have been difficult to compare because very often the concentrations of the individual PPs in extracts were unknown. Second, certain experiments used the PPs obtained from different grapevine organs, which contain very heterogeneous levels of these molecules, as it occurs, for example, for leaves compared to other organs of the grapevine [6,7]. Third, starting from different sources of PPs (wines) diverse extraction methods were reported to display very different degrees of efficiency, and to produce chemically heterogeneous extracts when the effects of three critical variables (sample volume, volume of each eluent, and solvent percentage in eluent) were evaluated for non-polymeric phenol and tannin recoveries from wine [8]. This is a very important issue to be considered when purifying PPs-containing samples for studying biological or health-related investigations as polymeric phenols are not absorbed by mammalian cells but are able to bind to and affect nearly any enzyme or receptor, producing irrelevant results [3]. Fourth, further complexity arose from the different results that researchers have reported in vivo or in vitro [6]. For instance, PPs displayed a very promising in vitro antioxidant capacity leading to the misconception that their cellular protection was mainly due to direct antioxidant scavenging [9]. Rather, studies with cellular and animal models demystified this concept, and now we know that the mode of action of PPs goes far beyond their antioxidant potential but also that their brain bioavailability is very limited [10]. Last, investigators have often focused their attention onto one single PP or group of chemically related molecules, thus making it quite difficult a sound comparison of the neuroprotective potential of individual PPs.

PPs of potential biomedical interest belong to several chemical families among which the most widely studied are stilbenes, anthocyanins, flavonols, and flavan-3-ols.

Stilbenes are a group of PPs that raised much interest in viticulture, as they are involved in the grapevine response to biotic stress, but also and even most in biology and health science. In preclinical neuroscience studies, resveratrol, the most widely investigated stilbene and the prototype of grapevine PPs studied so far, has been reported to have neuroprotective, antioxidant and anti-inflammatory, properties. Much of its purported benefits have been related to its ability to activate a family of proteins called sirtuins [11], but there is considerable conflict in the literature as regarding the true neuroprotective potential of these proteins [12]. The literature concerning the activation of sirtuin 1 (sirt1) by resveratrol is similarly controversial: the first studies indicated resveratrol as an activator of sirt1 [13], but subsequent experiments revealed that the stilbene-dependent activation of sirt1 was a technical artifact [14,15,16]. Resveratrol has also been touted as a treatment to slow physiological aging and age-related diseases including dementia and Alzheimer’s disease (AD) to eventually extend healthy lifespan. However, although resveratrol significantly extended lifespan in yeast, worms, and fruit flies [13,17], most studies reported no effect in mammals [18,19] and, very recently, the molecule was reported to exhibit biphasic dose-dependent effects acting as an antioxidant or as a pro-oxidant at low and high concentrations, respectively [20].

In the present study, we tested the death-combating effects of *trans*-resveratrol and polydatin, the glucose derivative of resveratrol, also termed piceid. Polydatin is generally predominant in white wines, whereas in red wines *Z-* and *E-*resveratrol are both quantitatively important [21]. Resveratrol is accumulated also in grapevine leaves, where its concentration was found to be up to ten times higher in organically-managed vines, respect to conventionally grown vines, and in both cases extracts were effective in reducing the lipid and protein damages induced by hydrogen peroxide in the rat brain [22]. In addition, two very recent studies have shown that polydatin protected SH-SY5Y neurons after rotenone treatment to model Parkinson’s disease [23] or from oxidative stress [24].

The efficacy of anthocyanins as potential therapeutic agents to combat neurodegeneration was tentatively linked to the different levels of hydroxylation in the B ring of the flavilyum ion, as it was suggested that the anthocyanins with a catechol moiety in their B ring could be more effective in neuroprotection compared to those devoid of catechol [25]. Specifically, the non-catechol pelargonidin 3-O-glucoside protected neurons from the oxidative stress elicited by glutamate but was ineffective against nitric oxide-induced apoptosis [26], whereas the catechol-structure cyanidin 3-O-glucoside (di-hydroxylated) was neuroprotective under both experimental conditions [27]. In addition, a combination of several anthocyanins was found to be protective against H_2_O_2_-induced oxidative stress in cultured human neuroblastoma cells [28] or C6 glial cells [29]. We here tested peonidin 3-O-glucoside (Pn-3OG) as the main representative of the anthocyanins in a few but locally very important varieties of *Vitis vinifera* [30] and malvidin 3-O-glucoside (Mv-3OG), quantitatively, without any doubt, the most important anthocyanin found in grape berries, musts and wines [31]. Another reason why we investigated the neuroprotective effect of Mv-3OG is that in vitro studies on SH-SY5Y human neuroblastoma cells [32] or C6 glial cells [29] have demonstrated protection against oxidative stress. However, the situation in vivo was quite different as the molecule overpassed the blood–brain barrier (BBB) in rats, but had no detectable effects in reducing the generation of the amyloid β (Aβ) peptides that are critical for the onset and progression of AD [33].

Flavonols are important PPs conferring to vegetal tissue high or very high antioxidant properties. They were protective in vitro against reactive oxygen species (ROS) challenge of SH-SY5Y neuroblastoma cells [32] and quercetin 3-O-glucuronide was capable to interfere with the generation of Aβ through the modulation of several different independent cellular mechanisms [33]. Moreover, quercetin 3-O-glucuronide significantly improved basal synaptic transmission in a hippocampal slice ex vivo preparation to model AD [33].

Based on these observations and on the fact that quercetin glycosides (glucoside + glucuronide) are, quantitatively, the most important flavonol in grape extracts and wines, we tested quercetin 3-O-glucoside as representative of the flavonols.

Monomeric grapevine flavan-3-ols include (+)-catechin with its diasteroisomer, (−)-epicatechin, and gallocatechin with its diasteroisomer epigallocatechin, differing for the level of hydroxylation in the B ring [34]. Monomeric flavan-3-ols were neuroprotective in a rat model of AD [35]. Therefore, we here have investigated (+)-catechin for its potential in limiting neuronal cell death.

Although at present not much is known about grapevine flavanonols, (+)-taxifolin (dihydroquercetin) has been very recently found to protect neurons against ischemic injury in vitro via the inhibition of excessive ROS production and of the irreversible increase of cytosolic Ca^2+^ concentration in GABAergic hippocampal neurons subjected to oxygen and glucose deprivation (OGD) to mimic ischemia [36].

Despite the notable amount of preclinical data on these four families of PPs, human clinical studies are occasional and thus the true translational potential of grapevine and other PPs in clinical neurology remains almost fully unexplored. For example, AD patients have lower cortical levels of sirt1, which indirectly correlated with greater levels of Aβ plaques and tau protein tangles [11,37]. Conversely, subjects with mild cognitive impairment (MCI) did not show reduced cortical sirt1 levels [11], indirectly suggesting that preventing sirt1 decreases at early stages of dementia may help delay or prevent the progression to AD. However, there is no evidence that treating humans with resveratrol can increase sirt1 in the brain [20]. In addition, clinical trials have up to now failed to show ameliorations of the clinical conditions in neurological patients under a resveratrol regimen [38,39] or epigallocathecin gallate [40] and a recent systematic review did not find sufficient evidence to confirm that PPs have beneficial effects against AD and other neurodegenerative conditions [41].

In the attempt to shed more light onto the neuroprotective potential of some grapevine PPs of shared interest for the viticulturists and the neuroscientists we have devised an initial study aiming to clarify the intervention of these molecules in protecting neurons from naturally occurring neuronal death (NOND) during the course of cerebellar maturation. Given the aforementioned limitations of much of the preclinical studies on the subject and the difficulty in translating these studies into the clinics, we have limited our work to a simplified and restricted experimental paradigm that: *i.* Exploits a well-known and widely characterized model of NOND [42]; *ii.* Compares the effects of commercially available purified PPs to overcome the problems that are inherent to the different extraction procedures; iii. Uses a slice culture approach to better mimic the in vivo situation; and iv. Uses a standardized concentration of PPs (50 µM) that is well below the highest doses (200 µM and above) employed in a wide number of studies in vitro [43]. We also discuss our results in relation to the information in the literature obtained from in vivo, ex vivo, and in vitro approaches.

## 2. Results

### 2.1. Effect of Ethanolic Media onto Naturally Occurring Neuronal Death (NOND) in Postnatal Cerebellum

During postnatal cerebellar development there is a well-characterized period of apoptotic cell death [42] currently referred to as naturally occurring neuronal death (NOND). NOND primarily affects the developing granule cells and is a massive phenomenon, so that it can be easily followed in slice cultures and is amenable to quantitative analysis (Figure 1A). As PPs are soluble in ethanolic solutions and ethanol itself induces death in neurons, we have devised a series of experiments in which cerebellar slices were maintained in vitro in the presence of progressively increasing concentrations of ethanol to assess the outcome on cerebellar NOND (Figure 1).

The graphs in Figure 2A,B show the effects of 1:100, 1:50, and 1:25 ethanol in medium (corresponding to 170, 340, and 680 mM, respectively) on the density of dead cells after PI staining (# cells stained with PI/mm^2^). Notably, the density of dead cells (mean ± 95% CI) raised from 17.56 ± 17.73 in plain medium, to 23.47 ± 8.76 (170 mM ethanol), 34.24 ± 2.30 (340 mM ethanol), and 78.11 ± 22.41 (680 mM ethanol).

Dispersion of data in 680 mM ethanol was very likely due to the severe toxic effect of the alcohol onto slices that, differently from the two other experimental conditions in the study, displayed obvious morphological signs of tissue sufferance such as fragmentation, disaggregation, vacuolization, etc. It is worth noting that in vitro experiments onto cultured primary cerebellar granule cells demonstrated that 25 mM ethanol was already inducing death, but alcohol was generally used at much higher concentration (87–200 mM) to reach better statistical significance [44,45]. Thus, the concentration of ethanol in our PP control media is within the range of these in vitro experiments and compatible with that in organotypically cultured cortical neurons [46]. Yet our experiments show that 34 and 68 mM ethanol produced statistically significant increases in NOND, whereas there were no differences in the mean density of dead cells at 0 and 170 mM ethanol (mean ± 95% CI: 17.56 ±1 7.73 (no ethanol), 23.47 ± 8.76 (170 mM ethanol), adjusted P value = 0.0815). We have also done a linear regression analysis to model the relationship between ethanol concentration in medium and cell death and found that the two variables showed a very high goodness of fit (R^2^= 0.9386, Figure 3). The Pearson’s correlation coefficient was r = 0.9688.

Based on these observations, we hold those experiments with PPs in which control- and PPs-supplemented media contained 170 mM ethanol allowed us to monitor NOND appropriately and not ethanol-induced death.

### 2.2. Effects of PPs onto NOND and Ethanol-Induced Cell Death

Figure 1 shows, as an example, the results of incubating the organotypic cultures in the presence of 50 µM resveratrol (Figure 1B) or 50 µM (+)-catechin (Figure 1C).

We have statistically tested the effects of PPs onto cerebellar NOND using two-tailed Mann–Whitney tests (Figure 4 and Figure 5). Statistics demonstrated that all PPs, except Mv-3OG (Figure 5G,H), were capable to reduce the density of dead cells in cerebellar cultures, with high significance.

We also observed that all PPs were as well effective in reducing ethanol-induced cell death (F = 15.11, P value < 0.0001) after Kruskal–Wallis test and Dunn’s multiple comparison test (P values for all comparisons against 680 mM ethanol as a control < 0.0001). Specifically, the mean density of dead cells per mm^2^ dropped from 78.11 (680 mM ethanol) to 16.20 (Pn-3OG), 13.59 (resveratrol), 10.53 (Q-3OG), 9.41 (Mv-3OG) 9.15 (Cat), 3.30 (polydatin), and 1.25 (taxifolin).

### 2.3. Comparison of the Effectiveness of PPs in Counteracting NOND

As we have related the effects of each PP against its own ethanolic (Figure 4A–C,E–H) or aqueous (Figure 4D) control, our experimental setup did not allow for correctly performing multiple comparison tests to make statistical inferences about the existence of possible differences in neuroprotective activities among the molecules used in this study. Nonetheless, we have calculated the ratios of the density of dead cells in controls and in the presence of each PPs (Figure 5A–C,E–H) and, thus, the per cent reduction of cell death for each molecule (Table 1). It is noteworthy that among the PPs here studied, the stilbenes (resveratrol and polydatin) appeared to be less effective in reducing cell death in comparison to Cat (a flavan-3-ol), Q-3OG (a flavonol), (+)-taxifolin (a flavanonol), and Pn-3OG (an anthocyanin). These four molecules, in fact, displayed very close percentages of reduction of cell death (69–72%) that were by far higher than those calculated for the stilbenes (51–60%). A remarkable observation was that, albeit in a statistically not significant way, aqueous Mv-3OG only reduced the density of death cells to 40% and even increased it to 127% in ethanolic solution.

## 3. Discussion

In this study, we have analyzed the neuroprotective effects of seven different PPs among those known to be present at higher concentration in grapevine.

Cell death in the postnatal cerebellum is a well-known physiological neurodevelopmental event mainly affecting the cerebellar granule cells that are the largest population of neurons in central nervous system (CNS). Being they so numerous, NOND of the granules is a massive phenomenon, and occurs in a quite restricted and tightly regulated temporal window [47]. Therefore, the use of postnatal cerebellar cortex organotypic cultures offers an adequate tool to study the neuroprotective potential of PPs in a controlled experimental setup [42].

Using this approach, we have demonstrated that all PPs studied here, except Mv-3OG, were capable to reduce NOND with statistical significance. We have also proved that all molecules without exceptions were also effective in counteracting the neurotoxic effects of 680 mM ethanol, a very high concentration in relation to studies on the toxic effects of alcohol.

We will first discuss our results in relation to the suitability of the ex vivo approach to investigate grapevine PPs neuroprotection, then we will briefly consider the PPs’ chemical structure and antioxidant activities and finally take into consideration the biological relevance of our findings.

### 3.1. Suitability of the Ex Vivo Approach to Study the Neuroprotective Effects of Grapevine PPs

As summarized in the Introduction, preclinical works aiming to characterize the biological and protective activity of PPs in the frame of neurodegeneration/inflammation display several limitations to the point that there is a strong debate onto their real translational relevance. We here used an ex vivo method to get rid of some of these limitations. Yet the use of organotypic cultures is not free of problems and does not represent a situation without controversies. We are aware of the shortcomings of our approach that paves the way for future better-focused pharmacological studies in vivo. Still, we believe it useful to discuss here the main problems related to the (generally) scarce bioavailability of PPs after in vivo administration. Taking resveratrol (and the stilbenes in more general terms) as the paradigmatic PP, its quantity in red wines is usually around 0.6 mg/L [21] but stilbenes can be up to 35 mg/L in certain Piedmont’s red wines and autochthonous Uvalino, in particular, contains up to 100 mg/L of resveratrol [48,49]. These figures correspond to concentrations of 2.6, 135, and 438 µM, respectively. Indeed, data are very dissimilar between studies and substantial differences exist in the reported concentrations of the several PPs that may be present in wines. It is thus remarkable that those of most PPs here studied (other than resveratrol) range in wines from 45 to above 750 µM [50,51]. To this, one must add that different wines have different profiles of PPs deriving from grape seeds, skins, and pulps [52]. However, one can reasonably conclude that in wine the molar contents of many of the PPs that we have studied, except for resveratrol but not in the autochthonous Piedmont Uvalino, are above those used ex vivo in our study.

Yet, the real issue is the bioavailability of PPs and, primarily, the brain concentration that they may reach in vivo. Indeed, Tomé-Carneiro et al. [43], in reviewing preclinical and clinical studies on resveratrol, evidenced that the former often used concentrations up to 200 µM but that resveratrol, quercetin, and catechin and their metabolites were scant in both plasma and urine (max 2 µM). Therefore, from these (and other) considerations they concluded that most studies in vitro were irrelevant.

Comparing our figures with those obtained from other preclinical surveys is not easy principally because, in most cases, results are expressed as the PP quantity in relation to brain weight, a very low quantity indeed, in the order of ng or even pg/mg of nervous tissue [53]. A more rigorous comparison takes into consideration the PPs concentration in the cerebrospinal fluid (CSF), which bathes the brain in vivo similarly to medium in our cultures. No resveratrol was detected in CSF after intravenous infusion in rat, but, after nasal delivery in chitosan-coated lipid microparticles, resveratrol reached a C_max_ after 60 min of 9.7 ± 1.9 μg/mL [54], corresponding to 34–51 µM.

Data onto the CSF content of PPs are scarce, but this paper strongly indicates that by choosing an appropriate pharmacological preparation as well as an efficient route of administration it is possible to achieve brain concentrations of resveratrol (and other PPs) comparable with those here used ex vivo. This is an important result supporting the relevance ex vivo approaches to study PPs neuroprotection.

### 3.2. The Relationship Between PPs Chemical Structure and Neuroprotective Effects

The increasing interest in PPs extracted from vegetal matrix in relation to neuroprotection is primarily due to their widely established antioxidant capacity (AOC) that, following several different cellular mechanisms and pathways, may be beneficial to neurons [6,9]. In turn, AOC depends on the chemical structure of individual PPs. In addition, the molecular composition influences the bioavailability of PPs, as different aglycones display distinct efficiencies in cellular transport/absorption [55]. In vivo, bioavailability refers to the fraction of a drug/molecule that, after absorption, reaches unchanged the systemic circulation. When dealing with CNS, drugs/molecules must be able to also cross the BBB for reaching the nervous tissue and thence exert their biological activities [56]. In a system ex vivo, BBB is not an issue, yet PPs need to be able to cross cell membranes to reach intracellular compartments and have some sort of efficacy. Other issues that are irrelevant to the present discussion, as we have at the moment tested each PP separately, is the possible synergy or antagonism among different families of PPs that are present in grapevine extracts [57] as well as the degree of polymerization of certain natural compounds, such as proanthocyanidins and copigments [58].

Among the PPs studied, we have here observed that the flavonoids were more effective than the stilbenes in protecting cells from death, except for Mv-3OG (see below).

Pn-3OG (anthocyanin) and Q-3OG (flavonol), both glucosides and di-substituted (Figure 6), showed a very high and statistically significant capacity to reduce NOND, of 72% and 69% respect to the corresponding control, and were among the most effective PPs in this study. We have here used these two PPs in glucoside form for three reasons. First, previous studies have reported that flavonoid glucosides can be absorbed as such without the need to be hydrolyzed to an aglycone [55]. Second, glucosides showed significantly higher transport efficiency than galactosides [55]. Third, they very likely enter the BBB as such, beside as glucuronide form, as demonstrated for anthocyanins [6]. Our results indicate that nor the glycosylation of the C ring of a di-substituted flavonoid form or its methylation (in 3′ of the B ring like in Pn-3OG respect to Q-3OG) negatively influenced the capability to reduce NOND. The anthocyanin Pn-3OG appeared to be the most effective in limiting not only NOND but also to counteract the adverse effects of high concentrations of ethanol in media. Therefore, it might be postulated that the B ring level of hydroxylation not alone but together with the total unsaturation of the C ring, typical of the anthocyanin molecules [59,60], are the key elements for explaining the capacity of Pn-3OG to drastically limit neuronal cell death in our ex vivo paradigm.

The flavonoids (+)-catechin and (+)-taxifolin are also among the PPs demonstrating the highest efficacy in reducing NOND. They are aglycones, but, like Pn-3OG and Q-3OG, di-hydroxylated, in the B ring (Figure 6).

(+)-taxifolin (flavanonol) and Q-3OG displayed similar capacities to limit NOND, 69 and 70% respectively. The two molecules share identical hydroxylation of the B ring but two different unsaturations in C (Figure 6). In addition, (+)-catechin, (+)-taxifolin, and Q-3OG have an ortho-hydroxyl in B, which is usually the initial target of antioxidants [61].

The null effect exerted by Mv-3OG on NOND but not on ethanol-induced cell-death could be ascribed to several factors, among which is the tendency of tri-hydroxylated anthocyanins to degrade in vitro more rapidly than mono and di-hydroxylated anthocyanins [29] and the limited efficiency in cell transport of glucosides respect to galactosides [55].

Thus, there may be several chemical features of the different flavonoids that, in theory, contribute to the neuroprotective effects observed in this study, but additional observations will be required to substantiate a significant correlation with some or all of them.

The two stilbenes, resveratrol, and its glycoside (polydatin, also called piceid) were undoubtedly less effective in counteracting NOND. Yet, resveratrol capacity to limit cell death was higher respect to that of the corresponding glucoside. One of the main antioxidant mechanisms of resveratrol is based on the presence of two -OH groups in the ring B [62]. Besides, in the comparison vs quercetin, quercetin has a further reactive -OH in position 3′, which is absent in resveratrol. Thus, the resveratrol chemical structure may justify a lower AOC respect to that of other PPs, particularly some of the flavonoids that we have tested here.

### 3.3. Clues for In Vivo Neuroprotection

We have here tested PPs in a controlled environment ex vivo. In such a setup, advantages mainly derive from the possibility to manipulate carefully and specifically the system according to the experimenter’s need, whereas disadvantages derive largely from the lack of information about bioavailability, metabolism and capability of crossing the BBB, i.e., the group of data that can be gathered only in vivo. Yet our approach permitted to provide an empirical demonstration that all PPs studied here were capable to cross cell membranes and to interfere with NOND. In addition, a great advantage in the specific use of postnatal cerebellar cultures is that cell death in this system is chiefly governed by the intracellular levels of caspase-3 (CASP3), one of the most important executioner caspases in apoptosis [47]. Although it is difficult to make direct comparisons due to the extreme variability of approaches, still one observes that all PPs that we have studied have been demonstrated to interact with CASP3 in vitro and/or in vivo not only in neurons but also in neuronal-like cells (Table 2). Remarkably, it should also be added that most of these molecules appeared to inhibit the activity of CASP3 after Aβ toxicity, a hallmark of AD [63].

## 4. Materials and Methods

### 4.1. PPs

We tested seven different PPs belonging to five main groups present in wine and grapevine extracts (Figure 6): two anthocyanins, peonidin 3-O-glucoside (Pn-3OG) and malvidin 3-O-glucoside (Mv-3OG), two stilbenes, resveratrol (Res) and resveratrol-3-O-β-D-glucopyranoside (polydatin), one flavonol [quercetin 3-O-glucoside (Q-3OG), purchased as quercetin 3-O-glucopyranoside], one flavan-3-ol, (+)-catechin (Cat) and one flavanonol (2R,3R)-dihydroquercetin [(+)-taxifolin]. All PPs were purchased from Extrasynthèses (Genay, France). Stock solutions (5 mM) were prepared in absolute ethanol and/or water (for Mv-3OG only). Given the pilot nature of this work and the heterogeneity of results arising from the different wine PPs extraction procedures (see Introduction), we decided not to devise experiments using mixtures of PPs as these mix should more properly been produced once their (relative) concentrations in the territorial wines of interest will be fully established.

### 4.2. Animals

In this study, we used twenty-five 7-day-old mice. All animal procedures obtained authorization by Italian Ministry of Health and the Bioethics Committee of the University of Turin and were carried out according to the guidelines and recommendations of the European Union (Directive 2010/63/UE) as implemented by current Italian regulations on animal welfare (DL n. 26-04/03/2014). We kept the number of mice to the minimum necessary for statistical significance and we made all efforts to minimize animal suffering during sacrifice. We wanted to gather new information about the biological function(s) of grapevine PPs in a more complex system than primary neuronal cultures/cell lines, where interactions between different cell types are lost. Therefore, mice were employed; yet the use of an approach ex vivo permitted a reduction in their number according to the 3Rs principles.

### 4.3. Preparation of Cerebellar Cultures

Mice were euthanized with an overdose of intraperitoneal sodium pentobarbital. The brain was quickly removed and placed in ice cooled Gey’s solution (Sigma Chemicals, St. Louis, MO) supplemented with glucose and antioxidants (for 500 mL: 50% glucose 4.8 mL, ascorbic acid 0.05 g, sodium pyruvate 0.1 g). The cerebellum was then isolated and immediately sectioned in 350 μm parasagittal slices with a McIlwain tissue chopper (Brinkmann Instruments, Westbury, NY), while submerged in a drop of cooled Gey’s solution. Three cerebellar slices were plated onto Millicell-CM inserts (Millipore, Billerica, MA). Each insert was subsequently placed inside a 35 mm Petri dish containing 1 mL of culture medium. Medium composition was 50% Eagle basal medium (BME, Sigma Chemicals, Merck, Darmstadt, Germany), 25% horse serum (Gibco^®^, Life Technologies™, Carlsbad, CA), 25% Hanks balanced salt solution (HBSS, Sigma Chemicals, Merck, Darmstadt, Germany), 0.5% glucose, 0.5% 200 mM L-glutamine, and 1% antibiotic/antimycotic solution. Cultures were incubated at 34 °C in 5% CO_2_ for 4 days in vitro (DIV) before being treated with PPs.

### 4.4. Preparation of PPs-Containing Media, Incubation of Cultures with PPs and Staining of Dead Cells

Except for Mv-3OG that could also be directly dissolved in water, all other PPs were not soluble in water and were added to the culture medium from ethanolic stock solutions. Culture media containing 50 μM of each of the PPs were prepared by dissolving 10 μL 100% ethanol stock solutions in 1 mL medium. The same volume of ethanol was added to control media. In the case of Mv-3OG we prepared culture media both from ethanolic and aqueous stocks.

As alcohol is known to be toxic to neurons, we devised an ad hoc series of experiments to ascertain the effects of ethanol onto neuronal survival at the concentration necessary to prepare the PPs containing media (170 mM–corresponding to 10 µL ethanol/1 mL medium). In these experiments, we also incubated some cerebellar cultures in 340 mM or 680 mM ethanol (i.e., 20 or 40 µL ethanol/1 mL medium).

At DIV 5 cerebellar cultures were subdivided in five groups and incubated into: i. Fresh plain medium (control for ethanol toxicity at 170 mM); ii. medium with 170 mM ethanol (control medium for PPs experiments); iii. Medium with 340 mM ethanol; iv. medium with 680 mM ethanol; and v. Medium containing 50 µM of each PP dissolved in ethanol (at a 170 mM final concentration). After 24 h, cultures were incubated for 10 min in medium containing 1.5 mM propidium iodide (PI) to visualize the dead cells. We decided to employ PI, a general marker of cell death rather than focusing onto a more specific assay because there are severe concerns about the specificity of action of several plant PPs onto animal cells [3]. In addition, the number of cellular mechanisms that have been put into play for explaining the numerous purported actions of resveratrol and other vegetal PPs is so high that virtually all the major cell pathways appeared to be involved [82], at times with opposite effects [20]. In addition, PI is widely used to stain dead cells as it is extruded from live cells with an intact membrane irrespective of the mechanism and type of death [83] and PI-based experiments thus offered prompt and, very importantly, comparable readouts of the effect of each PP onto NOND, independently from its mechanism(s) of action.

After three washes in plain medium, cultures were fixed for 60 min in 4% paraformaldehyde dissolved in 0.1 M phosphate-buffered saline (PBS) pH 7.4–7.6. They were then washed in PBS (2 × 10 min) and double-distilled water (2 × 5 min) and finally mounted in fluorescence-free medium (Vectashield^®^ Antifade Mounting Medium, Vector Laboratories, Burlingame, CA). Slices were then photographed using a Leica DM6000 wide-field fluorescence microscope (Leica Microsystems, Wetzlar, Germany) with a 20 × lens. For each slice, three randomly selected microscope fields (0.3084 mm^2^) were photographed at a resolution of 1392 × 1040 pixels and PI-stained nuclei were counted with the “Count Particles” function of the ImageJ software (NIH, Bethesda, MD, USA) in an interval of area size between 12.56 and 78.50 µm^2^ (corresponding to a particle diameter of 4–10 µm). Results were expressed as the number of PI-stained cells (nuclei)/mm^2^, i.e., density of PI-stained cells (nuclei).

### 4.5. Statistical Analysis

GraphPadPrism^®^7 (GraphPadSoftware, San Diego, CA, USA) was used for statistical analyses. These included I. Linear regression analysis and Pearson’s correlation test on the effects of different ethanol concentrations on the density of dead cells in cultures. II. D’Agostino and Pearson omnibus normality test to check for normally distributed data. III. Unpaired two-tailed Mann–Whitney test to make comparisons between two groups (control and PP-treated cultures).

Multiple comparisons for control experiments with different ethanol concentrations were made using the non-parametric Kruskal–Wallis test followed by Dunn’s multiple comparisons test.

## 5. Conclusions

In conclusion, we here demonstrate the usefulness of the ex vivo approach to study the neuroprotective potential of grapefruit PPs. Further studies will be required to confirm the relevance of our experiments in vivo considering the well-known problems regarding the bioavailability of these molecules following conventional pharmacological administration.

## Figures and Tables

**Figure 1 molecules-25-02925-f001:**
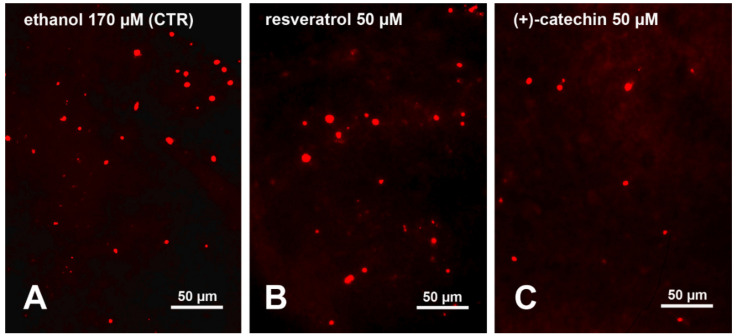
Limitation of naturally occurring neuronal death (NOND) in the postnatal cerebellum after ex vivo treatment with polyphenols (PPs). The nuclei of dead cells are strongly fluorescent in red after incubation with propidium iodide (PI). PI is a fluorescent intercalating DNA stain that is not membrane permeable. Thus, it only enters the nucleus of damaged cells and can therefore be used to differentiate dead cells (apoptotic, necrotic, etc.) from healthy cells based on membrane integrity. As 170 mM ethanol in which PPs are dissolved does not significantly alter cell death, it can be used as a baseline control (CTR) for the experiments to ascertain the effects of PPs onto NOND. The three panels are representative images of the experiments carried out with ethanol 170 mM (**A**) and with individual PPs, resveratrol 50 μM (**B**) and (+)-catechin 50 μM (**C**).

**Figure 2 molecules-25-02925-f002:**
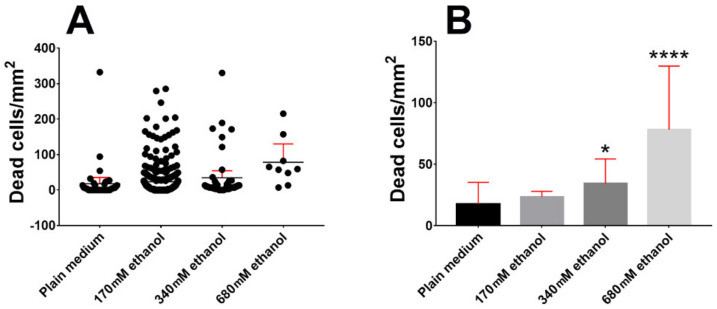
Descriptive exploratory statistics (**A**) and inferential statistics (**B**) of the effects of different ethanol concentrations in culture medium onto cerebellar naturally occurring neuronal death (NOND). Dead cells were stained with propidium iodide and results are expressed as means of dead cells/area. Two-tailed Kruskal–Wallis test and Dunn’s multiple comparison test (plain medium vs ethanolic media) were applied, as data did not pass the D’Agostino and Pearson normality test. * 0.05 > P > 0.01, **** P < 0.0001. Bars are 95% CI.

**Figure 3 molecules-25-02925-f003:**
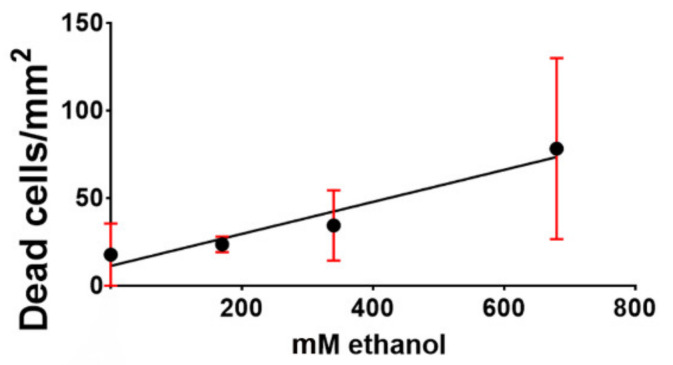
Linear regression curve of the density of dead cells related to different ethanol concentrations in culture media. Dead cells were stained with propidium iodide and results are expressed as means of dead cells/area. The curve demonstrates a positive correlation between the concentration of ethanol in medium and the density of dead cells. Slope was significantly non-zero: F = 30.55, P value = 0.0312. Equation: Y (dead cells/mm^2^) = 0.09141 × (ethanol concentration) + 11.15. Bars indicate 95% CI.

**Figure 4 molecules-25-02925-f004:**
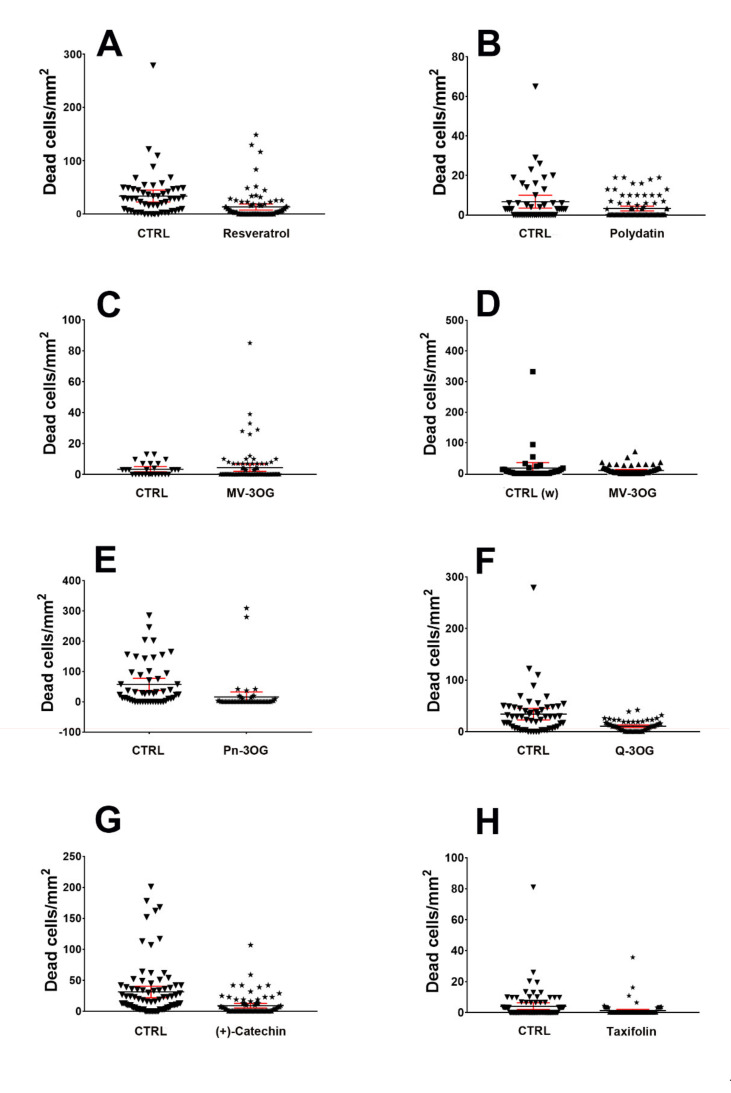
Descriptive exploratory statistics of the effect of different polyphenols (**A** = resveratrol; **B** = polydatin; **C** = Mv-3OG (malvidin 3-O-glucoside); **D** = aqueous medium-dissolved Mv-3OG; **E** =Pn-3OG (peonidin 3-O-glucoside); **F** = Q-3OG (quercetin 3-O-glucoside); **G** = (+)-catechin; **H** = taxifolin) on cerebellar naturally occurring neuronal death (NOND). Dead cells were stained with propidium iodide and results are expressed as means of dead cells/area with 95% CI. Scatter graphs show the dispersion and variability of data. Abbreviations: CTRL = control medium containing 170 mM ethanol; CTRL (w) = control medium (aqueous).

**Figure 5 molecules-25-02925-f005:**
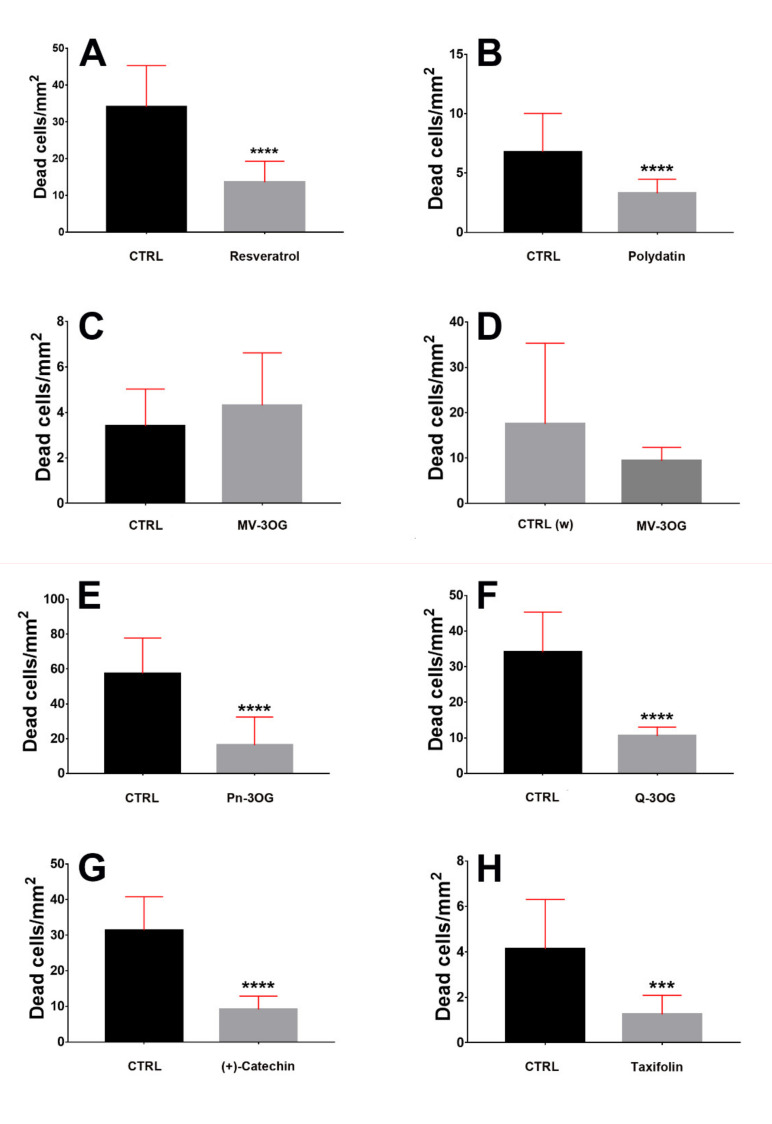
Inferential statistics of the effects of different polyphenols (**A** = resveratrol; **B** = polydatin; **C** = Mv-3OG (malvidin 3-O-glucoside); **D** = aqueous medium-dissolved Mv-3OG; **E** =Pn-3OG (peonidin 3-O-glucoside); **F** = Q-3OG (quercetin 3-O-glucoside); **G** = (+)-catechin; **H** = taxifolin) on cerebellar naturally occurring neuronal death (NOND). Dead cells were stained with propidium iodide and results are expressed as means of dead cells/area. The two-tailed Mann-Whitney test was applied, as data did not pass the D’Agostino and Pearson normality test. *** 0.0001 > P > 0.001, **** P < 0.0001. Bars are 95% CI. All PPs except Mv-3OG (C-D) reduced NOND in cerebellar slices. Abbreviations: CTRL = control medium containing 170 mM ethanol; CTRL (w) = control medium (aqueous).

**Figure 6 molecules-25-02925-f006:**
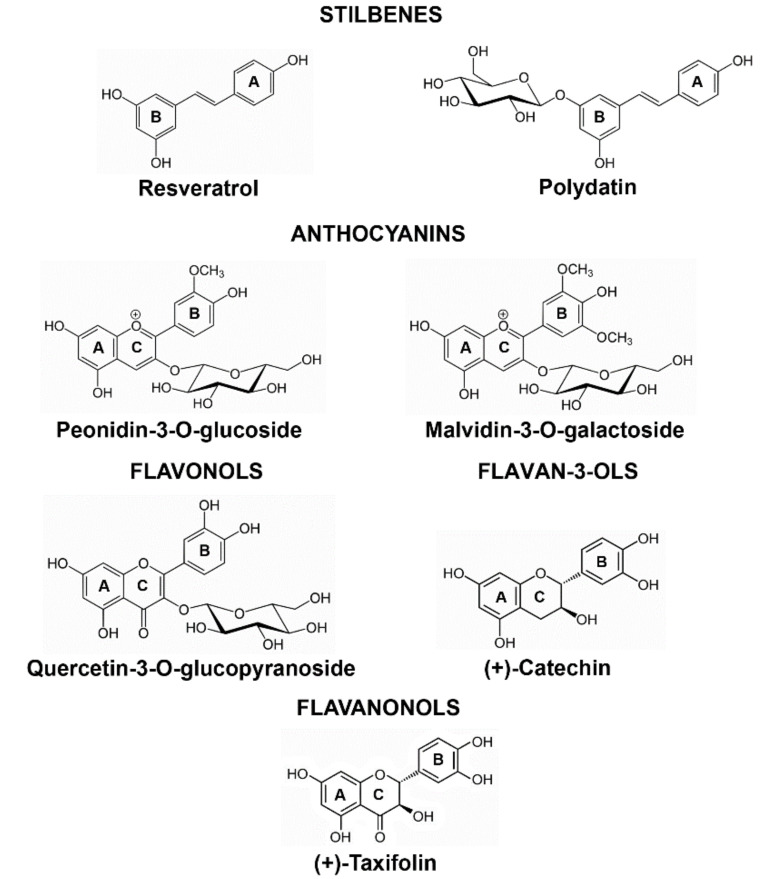
Chemical formulas of the PPs used to study neuroprotection in postnatal cerebellum.

**Table 1 molecules-25-02925-t001:** Percentages of reduction of the density of dead cells in controls (**a**) and after incubation in PPs (**b**) containing media.

PPs	Density of Dead Cells (PI^+^ cells/mm^2^)		Ratio (b/a)
	Control (a)	PP (b)	(c)
**Pn-3OG**	57.43	16.2	0.28
**(+)-Catechin**	31.32	9.15	0.29
**Taxifolin**	4.128	1.249	0.30
**Q-3OG**	34.04	10.53	0.31
**Resveratrol**	34.04	13.59	0.40
**Polydatin**	6.78	3.298	0.49
**Mv-3OG (Water)**	17.56	9.41	0.54

**Table 2 molecules-25-02925-t002:** Experimental studies demonstrating an inhibition of caspase-3 (CASP3)-mediated apoptosis after treatment with plant PPs. Abbreviations: Aβ = amyloid-beta; APP = amyloid precursor protein; IRI = ischemia/reperfusion injury; OGD = oxygen/glucose deprivation; LPS = lipopolysaccharide; N/A = not/applicable; PS1 = presenilin 1.

PPs	Type of study	Organ/tissue/cell/Species	Death inductor	Ref
**Resveratrol**	In vitro	PC12 cells/Rat	Aβ	[64]
		Primary cortical neurons/Rat	Aβ	[65]
		661W photoreceptor cells/Mouse	Blue light	[66]
		SH-SY5Y cells/human	Ethanol	[67]
	In vivo	Brain/Rat	Ethanol	[67]
**Polydatin**	In vitro	PC12 cells/Rat	Aβ	[64]
	In vivo	Primary cortical neurons/rat	Aβ	[65]
		Rat models of Parkinson’s disease	Rotenone	[68]
**Anthocyanins mix)**	In vitro	Hippocampal HT22 cells/Mouse	Aβ	[69]
		RGC-5/Mouse	H_2_O_2_ or Tunicamycin	[70]
		Primary hippocampal neurons/Rat	Ethanol	[71]
	In vivo	APP/PS1 mouse model of AD	N/A	[69]
		Brain/Rat	LPS	[72]
		Hippocampus/Rat	Ethanol	[73]
**Peonidin**	In vitro	RGC-5/Mouse	Tunicamycin	[70]
**Malvidin**	In vitro	661W photoreceptor cells/mouse	Blue light	[66]
**Quercetin**	In vitro	Primary cortical neurons/Rat	Aβ	[74]
		Primary hippocampal neurons/Rat	OGD	[75]
	In vivo	Brain/Rat	IRI	[75]
		Hippocampus/Mouse	IRI	[76]
		Brain/Rat	IRI	[77]
**Epicatechin**	In vivo	Brain/Rat	LPS	[78]
	In vitro	Auditory cells	Cisplatin	[79]
**Epigallocatechin**	In vitro	SH-SY5Y cells/human	Aβ	[80]
	In vivo	APP/PS1 mouse model of AD	Tunicamycin or Tapsigargin	[80]
**Taxifolin**	In vitro	PC12 cells/Rat	Proteasome inhibition	[81]

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
