# Peer review of "Protective Effects of Some Grapevine Polyphenols against Naturally Occurring Neuronal Death"

_molecules, 2020, doi:10.3390/molecules25122925_

Round 1
Reviewer 1 Report
Manuscript Number: molecules- 800329
Title Protective Effects of Some Grapevine Polyphenols against Naturally Occurring Neuronal Death
The study of natural neuroprotectors is very important since they can be used in the prevention and/or treatment of neurodegenerative diseases such Alzheimer Disease. Therefore, this study is of great interest. However, I found some problems.
Introduction
It is to long and some important information is missing. When the authors say that “results are often heterogeneous or inconsistent” (line 36), thy must exemplify the inconsistency with information found in literature. The same with the following affirmation.:
Lines 43– 49: Third, diverse extraction methods, often displaying very different degrees of efficiency, were employed 44 to produce the polyphenolic vegetal extracts in dissimilar experimental settings [5]. Fourth, further complexity arose from the different results that researchers have reported in vivo or in vitro [3]. Last, investigators have often focused their attention onto onea single PP or group of chemically related molecules, thus making it quite difficult a sound comparison of the neuroprotective potential of individual PPs.
In my opinion is much more interesting to explore the differences among the several studies that, according with authors, are contradictory that the long descriptions made in the Introduction about each polyphenol and that are not relevant to the paper. For example:
Lines 98-102: In grapevine organs, stilbenes can be constitutive, but their concentration in both vegetative and reproductive organs dramatically increases in response to stressful events, particularly, but not exclusively, of biotic origin. In grape berries, the glucose-derivative of resveratrol, also termed piceid or polydatin, is the prevalent stilbene, followed by Z- and E-resveratrol and/or viniferin [24].
Lines 121-132: They play a crucial role in enology that is fundamental as they react with anthocyanins and take part into co-pigmentation phenomena, thus contributing to wine colour stabilization [32, 33]. In plant biology, recent studies demonstrated that flavonols are involved in the plant response to water stress, dampering abscisic acid activity and favouring stomatal opening to control leaf gas–exchange [34]. In grapevine berry skins, the flavonol profile can be considered, at least partially, cultivar specific, e.g. myricetin glycosides are absent in white varieties, and the concentration of flavonols depends on environmental enlighten conditions. Rather than the individual molecules, it is instead the sum of the glycoside derivatives of a specific aglycone (i. e. quercetin or kaempferol) to be relatively cultivar- [35] or clone-dependent [36]. At harvest, berry skin flavonols typically include glucosides of quercetin and kaempferol rather than other types of glycosides, whereas glucuronides of quercetin and, to a lesser extent, of kaempferol, prevail in grapevine leaves [4, 37].
Lines 144-147: In grape berry skins, monomeric flavan-3-ols concentration more than fortyfold lower than in in seeds. The profile of monomeric flavan-3-ols is largely cultivar dependent [38] and (+)-catechin and (-)-epicatechin are generally the most abundant flavan-3-ols in ingredients derived from grape seeds. Catechin and epicatechin total antioxidant capacity is among the highest known, when measured by the TEAC method [39].
Material and Methods
Here is also where I found major problems. In the introduction the authors alert for the need of clarification about the grapevine polyphenols effects in the brain, pointing out several issues and then they do not present the solution to overcome the controversies. For example, in the Introduction the authors say that (lines 39-40) “First, researchers have sometimes tested the effects of pure molecules but other times those of crude plant extracts, with a substantial heterogeneity of results.” But in their study, they used “All PPs were purchased from Extrasynthèses 388 (Genay, France).” Why did not compare this with the natural extracts? Given the inconsistency pointed out by them they must explain the choice oh working with pure molecules. In the lines 47-49 the authors mention that “Last, investigators have often focused their attention onto one a single PP or group of chemically related molecules, thus making it quite difficult a sound comparison of the neuroprotective potential of individual PPs.” and in Material and Methods section line 421 “Culture media containing 50 μM of each of the PPs”. Why the authors did not try several PPs mixtures to explore their synergic effects? This must be clarified. In the Introduction the authors affirm “The link between wine consumption and "brain health" is further complicated because an excessive consumption of alcohol is, beyond any doubt, detrimental to the development and maturation of the nervous system [6, 7],” (Lines 53-55) but in their own study they use ethanol to solubilise PPs. Authors should explain several issues regarding the ethanol use. Compare the final concentrations of ethanol “170, 340 and 680 μM” to human wine consumption. Why they did not look for any other detrimental effect of ethanol in brain behind neuronal dead, i.e., could PPs counteract all the detrimental effects of ethanol? Is possible to discriminate naturally occurring neuronal death from the one caused by the ethanol? If not, could the authors discuss the PPs effects on NOND?
Why the authors did not carry out any experimental work to disclose the underlying mechanisms of PPs protection?
Results
The authors try to explore the few data that resulted from their experimental work and sometimes the text is unnecessary long.
Discussion
Too much speculation, as the authors did not carry out any experiment to disclose the mechanisms behind both neuronal dead caused by the ethanol and PPS protective effects, the authors speculated about them, specially comparing their chemical structure with their effects.
For the above I recommend that authors carry out additional work to address some the questions or change the text to better justify all their experimental choices. Discussion must be rewritten in a more pragmatic way and without speculation.

Author Response
Reviewer1
The study of natural neuroprotectors is very important since they can be used in the prevention and/or treatment of neurodegenerative diseases such Alzheimer Disease. Therefore, this study is of great interest. However, I found some problems.
We thank very much the reviewer for her/his appreciation of our work.
Introduction
It is too long, and some important information is missing. When the authors say that “results are often heterogeneous or inconsistent” (line 36), thy must exemplify the inconsistency with information found in literature. The same with the following affirmation:
Lines 43– 49: Third, diverse extraction methods, often displaying very different degrees of efficiency, were employed 44 to produce the polyphenolic vegetal extracts in dissimilar experimental settings [5]. Fourth, further complexity arose from the different results that researchers have reported in vivo or in vitro [3]. Last, investigators have often focused their attention onto one single PP or group of chemically related molecules, thus making it quite difficult a sound comparison of the neuroprotective potential of individual PPs.
In my opinion is much more interesting to explore the differences among the several studies that, according with authors, are contradictory that the long descriptions made in the Introduction about each polyphenol and that are not relevant to the paper. For example:
Lines 98-102: In grapevine organs, stilbenes can be constitutive, but their concentration in both vegetative and reproductive organs dramatically increases in response to stressful events, particularly, but not exclusively, of biotic origin. In grape berries, the glucose-derivative of resveratrol, also termed piceid or polydatin, is the prevalent stilbene, followed by Z- and E-resveratrol and/or viniferin [24].
Lines 121-132: They play a crucial role in enology that is fundamental as they react with anthocyanins and take part into co-pigmentation phenomena, thus contributing to wine colour stabilization [32, 33]. In plant biology, recent studies demonstrated that flavonols are involved in the plant response to water stress, dampering abscisic acid activity and favouring stomatal opening to control leaf gas–exchange [34]. In grapevine berry skins, the flavonol profile can be considered, at least partially, cultivar specific, e.g. myricetin glycosides are absent in white varieties, and the concentration of flavonols depends on environmental enlighten conditions. Rather than the individual molecules, it is instead the sum of the glycoside derivatives of a specific aglycone (i. e. quercetin or kaempferol) to be relatively cultivar- [35] or clone-dependent [36]. At harvest, berry skin flavonols typically include glucosides of quercetin and kaempferol rather than other types of glycosides, whereas glucuronides of quercetin and, to a lesser extent, of kaempferol, prevail in grapevine leaves [4, 37].
Lines 144-147: In grape berry skins, monomeric flavan-3-ols concentration more than fortyfold lower than in in seeds. The profile of monomeric flavan-3-ols is largely cultivar dependent [38] and (+)-catechin and (-)-epicatechin are generally the most abundant flavan-3-ols in ingredients derived from grape seeds. Catechin and epicatechin total antioxidant capacity is among the highest known, when measured by the TEAC method [39].
We have modified the text according to the reviewer’s suggestions and deleted all chemical description of the individual polyphenols. We also have modified the Introduction in response to Reviewer’s 2 specific comments on this section of the manuscript.
Material and Methods
Here is also where I found major problems. In the introduction the authors alert for the need of clarification about the grapevine polyphenols effects in the brain, pointing out several issues and then they do not present the solution to overcome the controversies. For example, in the Introduction the authors say that (lines 39-40) “First, researchers have sometimes tested the effects of pure molecules but other times those of crude plant extracts, with a substantial heterogeneity of results.” But in their study, they used “All PPs were purchased from Extrasynthèses 388 (Genay, France).” Why did not compare this with the natural extracts? Given the inconsistency pointed out by them they must explain the choice of working with pure molecules.
In the lines 47-49 the authors mention that “Last, investigators have often focused their attention onto one a single PP or group of chemically related molecules, thus making it quite difficult a sound comparison of the neuroprotective potential of individual PPs.” and in Material and Methods section line 421 “Culture media containing 50 μM of each of the PPs”. Why the authors did not try several PPs mixtures to explore their synergic effects? This must be clarified.
We thank the reviewer for this observation. We have clarified the reasons of our choice as follows: Given the pilot nature of this work and the heterogeneity of results arising from the different PPs extraction procedures from wine (see Introduction) we decided not to devise experiments using mixtures of PPs as these mix should more properly been produced once their (relative) concentrations in the territorial wines of interest will be fully established (lines 396-399 in Materials and Methods).
In the Introduction the authors affirm “The link between wine consumption and "brain health" is further complicated because an excessive consumption of alcohol is, beyond any doubt, detrimental to the development and maturation of the nervous system [6, 7],” (Lines 53-55) but in their own study they use ethanol to solubilize PPs. Authors should explain several issues regarding the ethanol use. Compare the final concentrations of ethanol “170, 340 and 680 μM” to human wine consumption. Why they did not look for any other detrimental effect of ethanol in brain behind neuronal dead, i.e., could PPs counteract all the detrimental effects of ethanol? Is possible to discriminate naturally occurring neuronal death from the one caused by the ethanol? If not, could the authors discuss the PPs effects on NOND?
We thank the reviewer also for this observation. First, it has been very important because, in trying to properly reply, we realized that we have erroneously indicated the ethanol concentrations in our media as µM instead that mM. Having said so, we apologize because, obviously, we have not been clear enough in explaining the rationale of using ethanol in our experiments. Indeed, we were not directly interested to assess the possible neuroprotective effects of the PPs against the ethanol-induced cell death, nor to analyze ethanol brain toxicity itself. However, ethanol was necessary to dissolve most of the molecules before adding them to culture medium. Therefore, in our experiments by using higher (340 and 680 mM) concentrations of ethanol than that (170 mM) necessary to dissolve the PPs we simply wanted to ascertain the effects of these concentrations of alcohol on naturally occurring neuronal death (NOND) to eventually exclude conditions in which ethanol was itself inducing death. These concentrations were thus totally unrelated to those that may be reached in the brain after human wine consumption, but rather they depended from the volume of the ethanolic PP solution that was added to culture medium. As this volume was usually 10 µL/1 mL (i.e. a final concentration of 170 mM in the medium) we wanted to confirm that we were analyzing the effects of each PP onto NOND and not onto ethanol-induced death. To make this better understandable, we have changed the legends in Fig. 4-5 graphs indicating the medium containing 170 mM ethanol as control medium.
We acknowledge that mentioning the French paradox on wine consumption in the Introduction of the paper was misleading and thus we have removed it in the revised MS. Moreover, we have tried to clarify our reasoning in the revised MS by grouping together Paragraphs 4.4. (Induction of cell death by ethanol) and 4.5. (PPs incubation and staining of dead cells) of the original MS in a revised 4.4. paragraph entitled “Preparation of PPs-containing media, incubation of cultures with PPs and staining of dead cells” (line 428). In the revision of the MS we have also tried to make our reasoning brighter. In addition, we have slightly changed the title of paragraph 2.1 that now reads “Effect of ethanolic media onto…” rather than “effect of ethanol… “ (line 155) to make sharper to the readers the rationale behind these experiments.
Coming to the ethanol concentration in our work, previous in vitro experiments onto cultured primary cerebellar granule cells demonstrated that 25 mM ethanol was already inducing death but alcohol was generally used at much higher concentration (from 87 to 200 mM) to reach better statistical significance (Pantazis et al J Neurochem 1998, 70:1826; Kouzoukas et al. Neurosci Lett 2018:108). In addition, studies in organotypic embryonic rat cortical cultures demonstrated that concentrations of ethanol in the range of about 40-80 mM for 3 days were sufficient to induce significant increases in the number of PI-stained cells (Mooney and Miller Dev Brain Res 2003 147:135). These studies indicate that organotypic cultures have a similar vulnerability to ethanol to that in vivo, whereas primary cultures are more resistant, and, in general, it is believed that ethanol concentration must be two- to three-fold greater in vitro to replicate the functional changes in vivo (see e.g. Luo and Miller Brain Res 1997 770:139; Jacobs and Miller J Neurocytol 2001 30:391). Thus, we wanted to be sure that the ethanol added to culture media was not altering significantly NOND in our experiments.
To better clarify these concepts, we have added the following sentences at lines 188-192 of the revised MS: “It is worth noting that in vitro experiments onto cultured primary cerebellar granule cells demonstrated that 25 mM ethanol was already inducing death but alcohol was generally used at much higher concentration (87-200 mM) to reach better statistical significance [44, 45]. Thus, the concentration of ethanol in our PP control media was within the range of these in vitro experiments and compatible with that in organotypically cultured cortical neurons [46]. Yet our experiments…”
Regarding the more specific issues raised by the reviewer:
- Compare the final concentrations of ethanol “170, 340 and 680 μM” to human wine consumption.
Regarding the comparison between the concentration of ethanol used in this study and that reached in humans after wine consumption, we would like to again emphasize that we have not designed our experiments to this purpose. Yet, the reviewer’s observation prompted us to examine the relevant literature on this very important issue. First, it is worth mentioning that most authors working on this subject have considered a dose of 87 mM ethanol physiologically high (e.g. Pantazis et al. 1998 J Neurochem 70:1826; Kouzoukas et al. 2018 Neurosci Lett 676:108), but, at the same time, reported that they were similar to those in the blood of chronic alcoholics (Minion et al. 1989 J Toxicol Clin Toxicol 27:375) and in rodent experiments on ethanol brain toxicity (see e.g. Bonthius et al. 2002 Brain Res Dev Brain Res 138:45; Ke et al. 2011 Alcohol Clin Exp Res 35:1574; Hubner et al. 2016 J Nutr 146:1180).
Yet a crucial issue, being ours a study ex vivo, would be to consider the concentration of ethanol in the brain after oral or parenteral administration. This is not trivial as ethanol is very rapidly removed from the nervous tissue by several enzymes, primarily alcohol dehydrogenase, and its brain concentration is, thus, technically very difficult to be measured in vivo (see Enrico and Diana, Front Behav Neurosci 2017 Vol 11 Article 97). Using micro dialysis, it was remarkably demonstrated that, after intraperitoneal administration of 1-4 g/kg ethanol to mice, both the blood and brain levels of alcohol reached similar peaks around 10-50 mM five minutes thereafter (Jamal et al. Neurochem Res 2016 41:1029). This indicated that the blood and brain levels of alcohol are very close within the first minutes of assumption in vivo. In keeping with these observations, one of us has recently shown (Ferrini et al Cell Mol Neurobiol 2018 38:955) that exposure of acute mouse cortical slices to 50 mM ethanol for 3-6 minutes was sufficient to elicit functional changes but did not induce cell death after iodidium propide (PI) assessment.
The doses of ethanol used by Jamal et al. in their study were chosen in relation to surveys on heavy drinkers and alcoholics and roughly correspond to an assumption from about 600 mL to 2.5 liters of wine (12% vol) by an 80 kg individual. As a glass (125 mL) of wine at 12° contains about 12 g ethanol whose specific weight (density) is 0.794, we can roughly deduce the following relation with the quantity of ethanol in our culture media:
|
Ethanol [] in media |
Ethanol absolute weight/L |
|
170 mM |
0.00794 g |
|
340 mM |
0.01588 g |
|
680 mM |
0.03176 g |
Thus, even at the highest concentration, the quantity of alcohol in our media is much lower than that assumed by a normal consumption of wine, but its molar concentration is quite high in relation to the blood and brain concentrations in vivo. These figures also confirm that our experiments, as devised, were not intended to study the neuroprotective effects of PPs in relation to wine consumption.
- Could PPs counteract all the detrimental effects of ethanol?
We could only speculate on this, as PI staining only allows for analyzing cell death based on the permeability of the cellular membrane. However, as we have tried to explain above, we were not directly interested in examining PP neuroprotection against alcohol, as also stated from the title of the MS.
- Is possible to discriminate naturally occurring neuronal death from the one caused by the ethanol? If not, could the authors discuss the PPs effects on NOND?
If we understand well, the reviewer argues that, not being possible to discriminate between NOND and ethanol-induced cell death, the discussion about the PPs effects on NOND is irrelevant. We respectfully disagree with this interpretation because our experiments with different concentrations of alcohol in media clearly show that the effect of 170 mM ethanol on NOND was negligible. We are confident that, as now revised, the MS better clarifies this issue.
Why the authors did not carry out any experimental work to disclose the underlying mechanisms of PPs protection?
Thank you very much for remark that gives us another possibility to explain better the rationale of our experiments. We have addressed this point at lines 445-453 of the revised MS.
Results
The authors try to explore the few data that resulted from their experimental work and sometimes the text is unnecessary long.
We have tried to shorten the results section as possible.
Discussion
Too much speculation, as the authors did not carry out any experiment to disclose the mechanisms behind both neuronal dead caused by the ethanol and PPS protective effects, the authors speculated about them, specially comparing their chemical structure with their effects.
For the above I recommend that authors carry out additional work to address some the questions or change the text to better justify all their experimental choices. Discussion must be rewritten in a more pragmatic way and without speculation.
We have condensed the text and better justified all our experimental choices.
Reviewer 2 Report
The article Molecules-800329 entitled "Protective effects of some grapevine polyphenols against naturally occurring neuronal death" is intended to explore the neuroprotective activity of polyphenols (authentic standards) in an ex vivo model with the aim of provided further pre-clinical evidence on the cellular mechanisms targeted by polyphenols in respect to neurological diseases. The article is well written and results presented, despite to be scarce for the ambitious objective of the article, are well discussed. Hence, the article still needs additional work to and diverse issues should be addressed before being published in MDPI-molecules.
Comments:
The aim of the article should be clearly stated in the abstract and in the last paragraph of the introduction. In the current version, the readers should guess the aim of the article during its lecture, and this makes understand results present and constitutes a serious constraint to draw conclusions.
On the other hand, the authors outline de limitations of the works available characterizing the biological activity of polyphenols in the frame of neuro-inflammation. I agree with this constraints regarding the difficult identification of the compounds responsible for the biological activity observed when working with whole extracts and the complementary information provided by alternative models testing the biological activity of individual compounds. However, this latter approach does not present a situation without controversies. In this regard, authors have to state clearely the restricted condition of the model chosen for testing the capacity of individual polyphenols to modulate neuro-inflammatory process, as well as the fact of using concentrations (50 microM) that never could be present in vivo in the target tissues. Thus, although this constitutes an interesting approach the results retrieved should be valorated carefully and jointly with additional information in the literature obtained by using additional in vivo, ex vivo, and in vitro approaches.
So, introduction should be simplified focusing in those issues addressed in the article resulting of the experimental work done. Otherwise, the article seems to be more a revision than an original contribution.
When describing results, the different figures are described in separate sections that difficult the comprehension of the graphics to begin with, although when continuing with the lecture, the description of figures in additional sections cover the lack of information detected. Thus, in order to facilitate the understanding of the text, the figure captions should provide a more complete information (abbreviations should be also defined in figure captions) that allows the figures stand by itselfs.
In page 5 170 micorM ethanol is associated to two distict SD (8.76 (first paragraph and 20.11 second paragraph). Clarify and correct if necessary.
In figure 3, remove the negative values of the “y” axis.
Why MV-3OG is the only compound tested in aqueous and ethanol solvents. Could be interesting to do the same approach for all compounds, especially given the clear differences displayed by this anthocyanin when using water or ethanol.
Table 1 should be simplified since 3 and 4 columns provide irrelevant information.
No determination of caspase 3 was done in the article. In this regard, the affirmation in the conclusions section seems speculative. This statement should be moved to the discussion and discussed resorting to information available in the literature and not referred in the conclusions section. In addition, the statement “…have a higher antioxidant activity, favoring…” should be modulated since neither SAR study was conducted in the article nor assessment of the radical scavenging activity of the compounds under evaluation.
Author Response
Reviewer 2
The article Molecules-800329 entitled "Protective effects of some grapevine polyphenols against naturally occurring neuronal death" is intended to explore the neuroprotective activity of polyphenols (authentic standards) in an ex vivo model with the aim of provided further pre-clinical evidence on the cellular mechanisms targeted by polyphenols in respect to neurological diseases. The article is well written and results presented, despite to be scarce for the ambitious objective of the article, are well discussed. Hence, the article still needs additional work to and diverse issues should be addressed before being published in MDPI-molecules.
We thank very much the reviewer for her/his appreciation of our work.
Comments:
The aim of the article should be clearly stated in the abstract and in the last paragraph of the introduction. In the current version, the readers should guess the aim of the article during its lecture, and this makes understand results present and constitutes a serious constraint to draw conclusions.
We have clearly stated the aim of the article in the abstract and last sentences of introduction.
On the other hand, the authors outline de limitations of the works available characterizing the biological activity of polyphenols in the frame of neuro-inflammation. I agree with these constraints regarding the difficult identification of the compounds responsible for the biological activity observed when working with whole extracts and the complementary information provided by alternative models testing the biological activity of individual compounds. However, this latter approach does not present a situation without controversies. In this regard, authors have to state clearly the restricted condition of the model chosen for testing the capacity of individual polyphenols to modulate neuro-inflammatory process, as well as the fact of using concentrations (50 microM) that never could be present in vivo in the target tissues. Thus, although this constitutes an interesting approach the results retrieved should be valorated carefully and jointly with additional information in the literature obtained by using additional in vivo, ex vivo, and in vitro approaches.
We warmly thank the reviewer for these observations that led us to consider better the controversies that are linked to the approach used in this study. Taking resveratrol (and the stilbenes in more general terms) as the paradigmatic PP, its quantity in red wines is usually around 0.6 mg/L (Guerrero et al. Nat Prod Commun 2009 4:635). Yet, stilbenes can be up to 35 mg/L in certain Piedmont’s red wines, particularly in Uvalino wines contains up to 100 mg/L resveratrol (Bertelli et al. Drugs Exp Clin Res 2004 30:111; Piano et al. Eur Food Res Technol 2013 237:897). These figures correspond to concentrations of 2.6, 135 and 438 µM, respectively. Indeed, figures may be very different according to the various authors and substantial differences exist among the several PPs that may be present in wines as e.g. summarized in the Table below:
|
PPs |
mg/La |
µM |
mg/Lb |
µM |
|
Catechin |
191 (R) 35 (W) |
657 (R) 120 (W) |
200 (Ry) 100 (Ra) |
689 (Ry) 344.5 (Ra) |
|
Malvidin |
24 (R) 1 (W) |
45.37 (R) 1.89 (W) |
400 (Ry) 90 (Ra) |
756 (Ry) 170 (Ra) |
|
Quercetin |
8 (R) 0 (W) |
17 (R) 0 (W) |
100 (Ry) 200 (Ra) |
215 (Ry) 430 (Ra) |
|
Resveratrol |
1.5 (R) 0 (W) |
6.5 (R) 0 (W) |
7 (Ry) 7(Ra) |
30.6 (Ry) 30.6(Ra) |
aFrankel et al J Agric Food Chem 1995 43:890; bCaruana et al. 2016 Front Nutr 3: Art 31 Abbreviations: R = red wine; Ra = red wine aged; Ry = red wine young; W = white wine
To this, one must add that different wines have different contents of PPs in grape seed, skin, and pulp (Pantelic et al 2016 Food Chem 211:243). Yet, one can conclude that the molar contents in wine of many of the PPs that we have studied except for resveratrol, but not in the autochthonous Piedmont’s wines, is well above that used in our study.
Nonetheless, the real issue, as correctly pointed out by the reviewer, comes as regarding the bioavailability of PPs and, primarily, their brain concentration in vivo. Indeed, Tomé-Carneiro et al (Curr Pharm Des, 2013, 19:6064) in reviewing the link between resveratrol and clinical trials e.g. observed that in vitro studies used resveratrol concentrations up to 200 µM but that the molecule and its metabolites as well as quercetin and catechin reached very low concentrations in plasma and urine (max 2 µM). Remarkably, from these (and other) observations it was concluded that most studies in vitro were irrelevant.
Comparing our data with those obtained from other preclinical studies is not easy principally because, in most cases, results are expressed as the PP quantity in relation to brain weight. Under this perspective the observation of the reviewer is very appropriate as there is a consensus that figures are very low, in the order of ng or even pg/g of nervous tissue (see e.g. Passamonti et al. 2005 J Agric Food Chem 53:7029), whereas concentrations of PPs in our experiments can be estimated to be much higher.
However, in our opinion, a more rigorous comparison should take into consideration the PPs concentration in the cerebrospinal fluid (CSF), which bathes the brain in vivo similarly to medium in our cultures. Under this perspective, it was recently and very interestingly observed that, whereas no resveratrol was detected in the rat CSF after an intravenous infusion, the nasal delivery of resveratrol in chitosan-coated lipid microparticles reached a Cmax after 60 min of 9.7 ± 1.9 μg/mL (Trotta et al 2018 Eur J Pharm Biopharm 127:250), corresponding to 34-51 µM.
To conclude, we have briefly discussed these issues in a new section of Discussion entitled: 3.1. Suitability of the ex vivo approach to study the neuroprotective effects of grapevine PPs at lines 275 and following of the revised MS.
So, introduction should be simplified focusing in those issues addressed in the article resulting of the experimental work done. Otherwise, the article seems to be more a revision than an original contribution.
We have modified the introduction following this observation, also in response to Reviewer’s 1 specific comments on this section of the manuscript.
When describing results, the different figures are described in separate sections that difficult the comprehension of the graphics to begin with, although when continuing with the lecture, the description of figures in additional sections cover the lack of information detected. Thus, in order to facilitate the understanding of the text, the figure captions should provide a more complete information (abbreviations should be also defined in figure captions) that allows the figures stand by itselfs.
We have modified the figure legends according to these suggestions.
In page 5 170 micorM ethanol is associated to two distict SD (8.76 (first paragraph and 20.11 second paragraph). Clarify and correct if necessary.
We apologize for the mistake and thank the reviewer for having spotted it. The correct value is 8.76 and we have corrected it in the second paragraph.
In figure 3, remove the negative values of the “y” axis.
Done. We have also removed the negative value of the Y axis in Fig. 4D.
Why MV-3OG is the only compound tested in aqueous and ethanol solvents. Could be interesting to do the same approach for all compounds, especially given the clear differences displayed by this anthocyanin when using water or ethanol.
We tested MV-3OG in both water and ethanol because the molecule was soluble in both solvents at the desired concentrations to prepare the stock solutions (100x - 5mM). However, other PPs could not be dissolved in water. We agree with the reviewer that it would have been interesting to also test all PPs in aqueous solutions, but it was necessary to prepare high concentration stocks for accuracy of the final dilution.
Table 1 should be simplified since 3 and 4 columns provide irrelevant information.
The table was modified as requested.
No determination of caspase 3 was done in the article. In this regard, the affirmation in the conclusions section seems speculative. This statement should be moved to the discussion and discussed resorting to information available in the literature and not referred in the conclusions section. In addition, the statement “…have a higher antioxidant activity, favoring…” should be modulated since neither SAR study was conducted in the article nor assessment of the radical scavenging activity of the compounds under evaluation.
We have rewritten the Conclusion as indicated. We have also modulated the statements about PPs antioxidant activity in Discussion also in response to Reviewer’s 1 comments.
Round 2
Reviewer 1 Report
The authors addressed the major concerns and the paper improved a lot, hoever introduction and discussion could be improved. After a carefully language review the paper will be acceptable for publication.
Reviewer 2 Report
The reviewed version of the MS addressed the concerns wake up by the previous draft. Although, some minor misspelling and grammatical errors were detected (that should be double-checked by the authors), in my view, the article is now ready to be accepted for publication.